# Refractory *Mycoplasma pneumoniae* Pneumonia in Children: Early Recognition and Management

**DOI:** 10.3390/jcm11102824

**Published:** 2022-05-17

**Authors:** Lin Tong, Shumin Huang, Chen Zheng, Yuanyuan Zhang, Zhimin Chen

**Affiliations:** 1Department of Pulmonology, Children’s Hospital, Zhejiang University School of Medicine, Hangzhou 310052, China; tonglin@zju.edu.cn (L.T.); huangsm93@zju.edu.cn (S.H.); aimianbao@zju.edu.cn (C.Z.); 2National Clinical Research Center for Child Health, National Children’s Regional Medical Center, Hangzhou 310052, China

**Keywords:** *Mycoplasma pneumoniae*, pneumonia, prediction, treatment

## Abstract

Refractory *Mycoplasma pneumoniae* pneumonia (RMPP) is a severe state of *M. pneumoniae* infection that has attracted increasing universal attention in recent years. The pathogenesis of RMPP remains unknown, but the excessive host immune responses as well as macrolide resistance of M. pneumoniae might play important roles in the development of RMPP. To improve the prognosis of RMPP, it is mandatory to recognize RMPP in the early stages, and the detection of macrolide-resistant MP, clinical unresponsiveness to macrolides and elevated proinflammatory cytokines might be clues. Timely and effective anti-mycoplasmal therapy and immunomodulating therapy are the main strategies for RMPP.

## 1. Definition and Manifestations

*Mycoplasma pneumoniae* (*M. pneumoniae*) is one of the most important pathogens for community-acquired pneumonia (CAP) in children. *M. pneumoniae* pneumonia (MPP) is typically mild and even presents as a self-limited disease. However, in recent years, more and more severe MPP (SMPP) [1,2] and refractory MPP (RMPP) [3,4,5,6] have been reported, posing great challenges to pediatricians. SMPP refers to the severe condition of MPP, which meets the criteria for severe CAP. Some SMPP patients even have a progressive process, developing rapidly into respiratory failure or life-threatening extrapulmonary complications, usually needing to be admitted to intensive care unit for life support and other treatment. Unlike SMPP, RMPP mainly indicates the difficult-to-treat conditions of MPP. This article provides a systematic introduction to RMPP in terms of definition, pathogenesis, clinical manifestations, predictions and management in the hope of providing some experience for clinicians and scientists.

*M. pneumoniae* is classically referred to as an “atypical” pathogen. Despite asymptomatically carriage having been reported [7,8], the pathogen can cause respiratory tract infections, such as tracheobronchitis [9], and in 10–40% of patients suffering from *M. pneumoniae,* infection will eventually develop into pneumonia [10,11,12,13,14]. MPP has long been considered to be the most frequent pathogen among school-aged children [15,16,17], but there has also been an increase in reports among preschool children and infants in recent years [18]. Although MPP is traditionally thought to be benign and self-limited, 18% of these patients require hospitalization [19], and some of them even progress into severe and fulminant or difficult-to-treat diseases.

The most common clinical manifestations of MPP include dry cough and fever, usually accompanied by headache, myalgias, sore throat, and abnormal findings on laboratory tests (mainly elevated inflammatory markers) and radiologic examinations (could be patchy airspace consolidation or just innocent in the X-ray, but bronchial wall thickening, centrilobular nodules, lymphadenopathy or ground-glass attenuation, among others, on chest computed tomography [20]) [9]. RMPP is usually used to describe the difficult-to-treat state of MPP, but it is not yet well defined. There are two main points which can be used to define a refractory case: (i) prolonged or even exacerbated clinical or radiographic manifestations; and (ii) unresponsiveness to appropriate treatment [21]. Until now, the most frequently cited definition is the criterion for inclusion in a case study reported in 2008 by Tamura et al., that is, a case of prolonged fever and deterioration of clinical and radiological findings after reasonable antimicrobial therapy for 7 days or more [3]. According to the latest guidelines for CAP in China, RMPP should be considered for patients who have been on macrolides for 7 days or longer, but still have aggravated clinical signs, persistent fever, and progressive pulmonary imaging [4]. It is often confused with the concept of severe *M. pneumoniae* pneumonia (SMPP), which focuses more on the severity of the disease, usually with ICU containment as a criterion [22]. RMPP, in contrast, has a considerably longer duration of fever, length of hospitalization, and greater likelihood of extra-pulmonary complications such as, but not limited to, pleural effusion and multi-organ dysfunction, and may also lead to more severe long-term sequelae, including bronchiolitis obliterans and bronchiectasis, among others. [23,24,25,26,27,28,29]. In recent years, more and more cases of RMPP have been reported in China and around the world [3,4,5,6], thus special attention is needed.

## 2. Pathogenesis

### 2.1. Pathogeny and Host Defense

*M. pneumoniae* can be transmitted through air droplets via coughing, sneezing and close contact. Vertical transmission has occasionally been stated in recent years [30,31]. The incubation period varies from 1 to 3 weeks, and the survival of *M. pneumoniae* in aerosols is thought to be related to meteorological conditions, especially humidity and temperature, but controversy still remains [17,32,33]. Once infected with the host, *M. pneumoniae* mainly adheres to ciliary cells of the mucosal epithelium, and close contact and material exchange between the bacterial membrane and the host cell provide an important material basis for its growth and proliferation. Bacterial cellular components such as glycolipids and capsular polysaccharides [34], virulence factors such as community-acquired respiratory distress syndrome (CARDS) toxin [35] and hydrogen sulfide, alanine, and pyruvate producing enzyme (HapE) [36], toxic metabolites such as hydrogen peroxide [37] and H_2_S [38], and nuclease [39], among others, are the main mechanisms for tissue damage. They also inhibit host clearance and promote immune escape [40].

CARDS toxin was first demonstrated in 2005 [41]. With a high sequence homology to the pertussis toxin S1 subunit, which performs ADP ribosylation and causes vacuolation, choristosis and spallation of mucosal cells. This toxin brings out the typical clinical symptoms of *M. pneumoniae* infection, for instance, dry cough or even spasmodic cough [42,43,44]. By other means, expressed CARDS toxin can also enhance the induction of the proinflammatory cytokines and stimulate lymphocyte activation in a dose- and activity-dependent manner [35,45,46,47] and is also capable of changing asthma-associated immunological parameters or inducing an allergic-type inflammation [40,48,49], potentially inducing or worsening asthma [7,50].

Hemolytic activity was also one of the identified pathogenicity determinants of *M. pneumoniae* where both hydrogen peroxide (H_2_O_2_) [51] and hydrogen sulfide (H_2_S) [52] contribute. Hydrogen peroxide is a metabolite of the process of glycerol utilization by Mycoplasma pneumoniae, and glycerol-3-phosphate oxidase (GlpO) is the key enzyme [53]. H_2_O_2_ is responsible for the oxidation of heme molecules and is also associated with oxidative stress and cell death [54,55]. H_2_S, as a by-product of the reaction to desulfurization of the cys by the enzyme HapE, can cause the modification of the heme and is responsible for the lysis of erythrocytes. By other means, H_2_S can also induce phagocytes to secrete pro-inflammatory factors, aggravating inflammatory reactions and leading to tissue damage [36].

Although the mechanism of RMPP is largely unknown, it has long been believed that the excessive host immune response plays a pivotal role in the disease progression [26,27]. Three mainstream hypotheses to explain the hyperimmune response for MP are summarized below [56]: (i) repeated or recurrent MP infections; (ii) loss of capacity to clear *M. pneumoniae* from the lungs in primary infection such as macrolide-resistance, which will be discussed later, resulting in a persistent MP infection; and (iii) an overactive innate immune response, such as macrophage activation through heterodimerization of Toll-like receptors [57,58]. The overall result of the above factors is an excessive and overactive immune response, which will be explained in detail in later parts.

### 2.2. Macrolide-Resistant M. pneumoniae (MRMP)

The lack of a cell wall renders *M. pneumoniae* intrinsically resistant to some antimicrobials, such as beta-lactams, glycopeptides and fosfomycin antimicrobials, which lays a trap for the identification of this atypical pathogen and also results in difficulties in treating pediatric *M. pneumoniae* infection. Historically, the main efficient drugs against *M. pneumoniae* include agents targeting the bacterial ribosome for inhibiting protein synthesis, such as macrolides, and others inhibiting DNA replication, such as fluoroquinolones [3]. Macrolides are the first and nearly the only choice for pediatric patients due to toxicity and side effects of other drugs for young children. Unsurprisingly, under the long-term pressure of antibiotic selection, macrolide-resistance emerged.

Thus far, the vast majority of reports correlating with macrolide-resistant infections were from children, due to a high incidence of *M. pneumoniae* infections and also the wide use of macrolides in pediatric age groups, but macrolide-resistance can also occur in adults [57,58]. Up to now, no difference has been found in disease manifestations between pediatric patients and adults infected by MRMP. Resistance of *M. pneumoniae* to macrolides was first described in 2001 in Japan [59] and quickly swept across East Asia, wherein the resistance rates were found to be higher than 90% in some countries during the epidemic years [60,61]. Since that time, a progressive increase in incidence rates of MRMP strains was reported worldwide, although with a significant difference among countries [24,56,62]. It is also a common and disturbing problem in China, both in adults and in children [63,64].

Temporal studies suggest that the emergence of significant resistance to macrolides by *M. pneumoniae* takes precedence over the peak of *M. pneumonia* episodes [25]. Therefore, the activation of resistant strains may be one of the important causes of the MP outbreak. Additionally, some studies have illustrated that macrolide-resistance of *M. pneumoniae* may play an essential role in RMPP development and progression, given the limited sensitivity of MRMP to macrolides may result in higher bacterial load and excessive immune response [5,65,66,67]. The opposite point of view also exists, demonstrating that macrolide resistance may not be associated with the development of RMPP [29,68]. Therefore, the association between RMPP and increased macrolide-resistance requires further investigation.

### 2.3. Co-Infection

Co-infection in CAP is clinically common. Likewise, the dual existence of *M. pneumoniae* with other organisms is not rare in patients with respiratory syndromes, especially in children [69]. The rates of viral (human bocavirus, rhinovirus, respiratory syncytial virus, among others, respectively) or bacterial (Streptococcus pneumoniae, Hemophilus influenzae, Staphylococcus, among others, respectively) coinfection with *M. pneumoniae* in children were reported ranging from 8 to 60% [70,71,72,73,74]. Some reports revealed simultaneous laboratory-proven infections with both bacteria and viruses in addition to *M. pneumoniae* [70,73].

Although the contribution of these coexistent agents remains unclear, since healthy individuals may carry these opportunistic pathogens as well [75,76,77], coinfection with viruses and bacteria causes more severe diseases in pediatric patients, according to previous research [78,79]. In children with RMPP, Zhang et al., demonstrated that coinfection with viruses and bacteria resulted in more severe processes [73]. Zhou et.al recently reported that adenovirus coinfection with MRMP was shown to be more prevalent in RMPP patients [66]. However, Chiu et al., found no significant difference in clinical features, complications, or outcomes between the patients infected with *M. pneumoniae* alone or with virus coinfection, despite the latter having prolonged fever and hospital stay [72].

## 3. Prediction and Early Recognition

Almost all previous reports indicated that delayed appropriate treatment was associated with the development of more severe and/or extended illnesses [9]. Thus, clinical awareness, prompt detection of *M. pneumoniae* and its macrolide resistance and early recognition of RMPP enable effective therapy to begin sooner, potentially improving clinical outcomes [68].

### 3.1. Clinical Awareness and Confirmation of Macrolide Resistance

The gold standard for diagnosing MRMP is culture and drug sensitivity. However, culture is much too time-consuming, thus the identification of MRMP strains is usually made with molecular biology methods nowadays. *M. pneumoniae* carry a total of 816,394 bp base pairs with 687 genes on the circulating double strands of DNA function to maintain their viability and reproduction [80]. Molecular epidemiology investigations of *M. pneumoniae* and macrolide susceptibility have been conducted in a wide range of geographical and temporal contexts [24,81,82,83,84]. Most investigations showed that MRMP usually had specific point mutations in the peptidyl transferase loop of 23S rRNA, as well as insertions or deletions in ribosomal proteins L4 and L22 [9]. Genotyping analysis from Japan suggested that epidemics arise due to variants of P1 sequences [40] and was further verified and refined in subsequent studies [41]. In China, variants in domain V of the 23S rRNA gene are also the major cause of MRMP, with most strains harboring an A2063G mutation, in which P1 type 1 and type 2 lineages co-circulate [29,63,85,86,87]. At present, commercial PCR kits for the rapid detection of both MP gene or antigen and drug resistance mutations simultaneously are available on the market [88,89,90], and makes it possible to rapidly diagnose MRMP.

Some clinical phenomena may also serve as early indicators of macrolide resistance MPP (MRMPP), especially macrolide unresponsiveness. Patients with MRMPP usually have an extended period of fever in spite of macrolide therapy. They are also more susceptible to more severe phenotype, and more complications [91,92]. For the early recognition and confirmation of MRMPP, pediatricians should pay more attention to the initial response to macrolide. If a child with confirmed or suspected MPP does not respond to macrolide therapy in the first three days (macrolide unresponsive MPP), MRMPP should be suspected and further management should be adopted, especially in countries and regions with high MRMP rates [93,94]. Coinfection with bacteria or virus and complications should also be excluded.

### 3.2. Early Identification of RMPP Cued by Cytokine Profiles

The host immune response is a “double-edged sword”. On the one hand, an adequate immune response including cytokine secretion and lymphocyte activation is essential for the elimination of *M. pneumoniae*, helping alleviate disease [95]. Children with hypogammaglobulinemia appeared to be more vulnerable to invasive and prolonged bacterial infections [96]. On the other hand, an improper immune response to *M. pneumoniae* generates excessive inflammation, and can exacerbate the disease clinically, even leading to the development of RMPP. Evidence revealed pulmonary lesions were generally mild in immunodeficient children [97]. This theory may also partially explain the selectivity of RMPP in terms of children’s ages. Children over the age of 5 years old have a relatively better developed immune system than younger children; coincidentally, the former group happens to be more susceptible to disease and exhibits more severe phenotypes of disease [5].

Although the direct correlation between the host immune response and RMPP is inconclusive, a growing body of evidence points to it. The course and outcome of mycoplasmal infection seem to be highly dependent on host responses. The stronger the immunological response and activation of cytokine, the more severe the clinical disease and organ damage. Herein, we wonder whether cytokine profiling may predict the severity and subtype of illness in advance, allowing for reasonable and individualized therapy adjustments to be made as early as possible.

Numerous literatures have reported the correlation between cytokines, chemokines or other inflammatory biomarkers and RMPP. Lactate dehydrogenase (LDH), for example, has long been regarded as a reliable evaluation index of RMPP. The cut-off value of LDH for considering RMPP ranged from 379 to 480 IU/L among adolescents and adults [94,98,99,100,101]. Our previous study suggested LDH ≥ 417 IU/L to be significant predictors in regard to RMPP [98]. Some other inflammatory biomarkers, such as CRP ≥ 16.5 mg/L [98], ESR ≥ 32.5 IU/L [99] and 35 α-hydroxybutyrate dehydrogenase (HBDH) ≥ 259.5 IU/L [99] also have indicative significance for RMPP in children.

To combat MP infection, neutrophils, CD8+ T cells, as well as Th1 biased CD4+ T cells, are recruited followed by enhanced humoral immunity. In recent years, more attention has been paid to proinflammatory cytokines. In our previous study, the percentage of neutrophils and CD8+ T cells, as well as the levels of IL-6, IL-10 and IFN-γ, were shown to be beneficial for distinguishing patients with RMPP from those with general MPP [98,102], serum chemokines such as CXCL10/IP-10 may also be potential biomarkers [103]. This phenomenon has been confirmed by other studies in recent years. Therefore, we should be alert to the possibility of RMPP when cytokines such as IFN-γ [5,83,102], TNF-α [5], IL-6 [83,98], IL-10 [102], IL-18 [104,105,106], among others, are obviously elevated. Further confirmation of these candidates is needed.

## 4. Management

### 4.1. Macrolides and Alternative Antibiotics

Macrolides represented by azithromycin and clarithromycin have long been the first-line antibiotics against *M. pneumoniae* due to their efficacy, safety and good tolerability, particularly in children [22,107,108,109,110]. As predicted, macrolide effectiveness was decreased in patients infected with macrolide-resistant isolates, resulting in intractable clinical symptom signs and laboratory examinations such as longer febrile day, prolonged hospitalization and antibiotic therapy, increased coughing and worse chest roentgenogram findings, among others. However, macrolides may be therapeutically beneficial in certain individuals infected with macrolide-resistant strains [111,112,113]. This finding can be partly explained by the fact that *M. pneumoniae* infections are often self-limiting and that the anti-inflammatory actions of macrolides may ameliorate clinical symptoms [114].

Nevertheless, a change in antibiotic prescription should be considered when symptoms persist despite the macrolide resistance, especially when RMPP is highly suspected. Clinical attempts on tetracyclines are uplifting and promising. Minocycline and doxycycline (recommended as twice-daily dose of 4 mg/kg/day [115,116]) were mostly reported with rapid effectiveness and a relatively low incidence of adverse reactions [94,113,116,117,118,119]. Doxycycline has also been recommended as the first alternate antibiotic for macrolide-resistant MPP [113]. Tetracyclines, on the other hand, are associated with depressed bone development, tooth enamel hypoplasia and permanent tooth discoloration in young children and should be prescribed only to children over the age of eight [120]. Another choice is fluoroquinolones, represented by tosufloxacin [118,121,122], moxifloxacin [123], ciprofloxacin [124] and levofloxacin [115,125,126]. The main considerations for fluoroquinolones are fears of articular problems and reversible musculoskeletal events [127]. In Japan, oral tosufloxacin was authorized as a second-line treatment for pediatric patients with CAP [56] at a dosage of 12 mg/kg/day [116]. The most common side effects are mild diarrhea, with seldom joint symptoms reported [118,121]. However, although its efficacy and safety have also been reported [124], the experience of fluoroquinolone administration in MPP children is rare.

Recently, newly investigated antimicrobial agents, such as Lefamulin [128,129], Solithromycin [129], Omadacycline [130] have received a great deal of attention. However, they are still far from clinical use, particularly in children [9].

### 4.2. Immunomodulating Therapy

Although corticosteroid medication is normally avoided in infectious diseases, it is helpful in RMPP patients, given the pathogenesis of RMPP is likely Immuno-mediated, at least in some parts, both in terms of lung damage and extrapulmonary involvement [131]. Several studies have examined the efficacy of systemic corticosteroids in children with RMPP, and it was shown that combining macrolide with corticosteroids was a superior treatment choice for children with RMPP than using macrolide alone, especially for *M. pneumoniae* extrapulmonary symptoms [94]. However, the most appropriate application timing and optimal protocol of corticosteroids remains unclear.

Corticosteroids, most commonly methylprednisolone or prednisolone, were used with a large variety in doses from 1 mg/kg/dose to 30 mg/kg/day with corresponding treatment duration and from oral take to intravenous injection. A retrospective study in Japan demonstrated that patients with RMPP treated with a regular dosage of corticosteroid (2 mg/kg/day or more of prednisolone for averagely 4 days before dose reduction) could achieve defervescence earlier and have a shorter hospitalization, when compared to the low-dose corticosteroid group (<2 mg/kg/day) [132,133]. A meta-analysis of relative articles from China demonstrated that high-dose (10–30 mg/kg/d) methylprednisolone was more efficient than a low-dose (1–2 mg/kg/d) strategy, both for 3 days without increasing the prevalence of adverse reactions [132,133]. Given that higher doses may be needed in patients with severe MPP, the pulse therapy (30 mg/kg/d of methylprednisolone usually for 3 days) could be applied in severe refractory MP and achieve good outcomes [134]. Considering appropriate timing, some authors claimed that early corticosteroid therapy, irrespective of used antibiotics, might reduce disease morbidity and prevent disease progression in MPP patients [135,136], as was observed in another prospective randomized clinical trial [137].

To avoid drug abuse, it is critical to search for an appropriate intervention point for initiating steroid treatment in MPP. Lots of investigations from China [6,100,124,132,138,139], Japan [3,101] and Korea [134,140] considered RMPP as an indication for corticosteroid application. However, the diagnosis of RMPP requires at least 7 days and more of macrolide antibiotic application as the basis, and too late corticosteroid intervention may lead to sustained inflammatory damage or even irreversible organ damage. Therefore, early intervention based on the recognition of excessive inflammatory response may be conducive to the early control of inflammation, preventing disease progression. Miyashita N et al., proposed that a serum LDH level of 302–364 IU/L seemed to be the threshold for starting corticosteroid treatment in severe or RMPP [21]. According to Oishi et al., an increasing IL-18 value (≥1000 pg/mL) was also indicative [141].

Most patients achieved defervescence within 72 h after the regimen of intravenous methylprednisolone of 2 mg/kg/day and no return of fever for at least 7 days after corticosteroid, except for approximately 20% of corticosteroid-resistant cases [138], whose fever may persist for more than 3 days following steroid treatment. In this case, increasing the dose of steroids or administering intravenous immunoglobulin (IVIG) could be considered [123,138,139]. Some authors advocated intravenous methylprednisolone at 4 mg/kg/day, followed by an increase to 6 mg/kg/day if fever persists [139]. Furthermore, IVIG might be indicative for those with neurologic impediments [142,143] or with uncontrollable rash and mucositis [144,145]. Still, to arrive at the optimal regimen and to draw sufficiently credible conclusions, more clinical studies are urgently needed [94].

### 4.3. Flexible Bronchoscopy

Flexible bronchoscopy has been well developed and has now become an important means of diagnosis and treatment for refractory respiratory diseases [146,147]. Despite being an invasive technique, it is still applied in some patients with pneumonia, particularly severe or difficult-to-treat patients, not only for identifying infectious agents or assessing the structure of lower airways, but also for therapy.

The benefits of airway plug removal and bronchoalveolar lavage (BAL) under bronchoscopy in pneumonia with persistent atelectasis have been indicated in several studies [148,149]. Disease remission was significantly accelerated after therapeutic bronchoscopic intervention, including clinical symptoms and laboratory findings, as well as resolution of atelectasis [150]. Early application of bronchoscopy might be more beneficial [151]. A prospective observation by Geng et al. [150] and a retrospective study by Zhuo et al. [152] also reached similar conclusions. There was no obvious adverse event in bronchoscopy procedure reported in studies. However, given the self-limiting nature of *M. pneumoniae* infection and the invasiveness of bronchoscopy, it is mandatory to balance the benefits and risks before adoption. Its indications must be strictly limited only in severe and persistent airway obstruction, such as persistent atelectasis and plastic bronchitis.

## 5. Conclusions

RMPP has received more and more attention in recent years, both due to the rising incidence and the difficulties it brings to treatment. Until now, its pathogenesis has remained unclear, but MRMP and excessive immune reaction might be the main reasons. Due to the lack of cell walls, MRMP leaves few options for the selection of antibiotics, posing a great challenge to pediatricians due to safety concerns regarding other sensitive antibiotics for young children such as tetracyclines and fluoroquinolones. More studies about the safety of these alternative antibiotics and the development of new drugs for MP are urgently needed. For excessive host responses, the immunoregulating agents corticosteroids and IVIG have both demonstrated potential benefits. Early intervention has shown to be more beneficial for reducing lung damage. However, the optimal time and regime of immunoregulating therapy are still obscure. The investigation of reliable predictors may help in the early identification of RMPP and provide the possibility for early intervention and finally improve the prognosis.

## Data Availability

Not applicable.

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
