# Peer review of "Refractory Mycoplasma pneumoniae Pneumonia in Children: Early Recognition and Management"

_jcm, 2022, doi:10.3390/jcm11102824_

Round 1
Reviewer 1 Report
sufficient summary regarding the current diagnostic and therapeutic status of M. pneumonie with some precautions to be carried out.
- insert a more detailed biological description of Mycoplasma specifying its distinctive peculiarity, and specifying the direct and indirect effects it has on the tissue line 55-58 eg. H2S as a by-product of the reaction to desulfurization of the cys by the enzyme HapE
- line 112-115 divide the percentages of viral and bacterial agents in the co-infection, specifying them if possible.
- line 129 insert which laboratory tests are altered and insert specific radiographic images of the MPP and RMPP, as an exemple.
- line 146 because it has not been specified that the laboratory diagnosis is mainly based on rapid culture of throat swab specimens, PCR and serological tests, such as ELISA, and in line 159 which commercial kit do they intend?
- pay attention to the missing brackets for references such as lines 123, 156.
Reviewer 2 Report
- General: The term Refractory Mycoplasma pneumoniae pneumonia (RMPP) is not recognized in the worldwide in the pulmonary community. Would be great to define this immediately in the introduction. This we call “Mycoplasma pneumonia with secondary complications” or better yet “secondary non-pulmonary complications”. How is this different from just severe mycoplasma? Is it due to extrapulmonary organ symptomatology? It is very unclear, and would be great to state this up front.
- Line 165-166 – this recommendation of no response on macrolide therapy on fever after 3 days is arbitrary, and lacks any scientific basis presented here. Either put a reference here from where this came from, or modify this to “no defervescence of fever in the first few days of treatment” – thus making it more ambiguous, allowing the treating physician to decide.
- Line 167 - MUMPP acronym is not necessary. Isn’t macrolide unresponsive MPP part of the refractory MPP? It is discussed as a subcategory. So woundn’t that be MURMPP? - that is just way too many unnecessary acronym productions. And what about MRMP (line 78)– macrolide resistance Mycoplasma pneumoniae pneumonia – would there be MRMPP not MRMP? That is way too close to RMPP, will need to make this acronym, if even needed, different here. Extremely
- Line 190 - LDH is a very nonspecific biomarker. It is not a reliable diagnostic of RMPP, due to the lack of specificity of elevated LDH. It is high in any disease state where cell death occurs – including cancer, etc. Did the authors mean the following? If so, please edit this sentence:
“lactate dehydrogenase, for example, has long been regarded as a reliable diagnostic of progression to RMPP in the setting of M. pneumoniae infection”
- Line 196 – similarly to above comment, it should state here “... indicative significance for RMPP in children with M. Pneumoniae infections. “ As all of those biomarkers are very nonspecific (CRP, ESR, etc), and can be elevated in any infection.
- Line 199-201 - what was the conclusion about neutrophils and CD8 T-cells in the previous study? And Il-6, IL-10 and IFn-gamma - were they elevated, decreased or how were they beneficial if distinguishing RMPP patients from uncomplicated MPP patients? Would suggest a quick synopsis about the findings here.
- Line 213-214 – RE: “However, macrolides may be therapeutically beneficial in certain individuals infected with macrolide-resistant strains”. This statement needs a justification – how may they be therapeutically beneficial (as decided by what criteria), and in what percent of cases – please tell us more about this. Faster recovery? Less ICU days? Less hospitalization days? What were the endpoints by which this conclusion can be made?
- Line 219 – remove MUMPP acronym (See comment 3 above)
- Line 237 – may need to add “particularly in children’ at the end of this sentence.
- Line 240 – need to soften this sentence, as this has not been proven to be the only reason there is RMPP. Should say “given the pathogenesis of RMPP is likely Immuno-mediated, at least in some parts, both in terms of lung damage and extrapulmonary involvement”
- General: in terms of corticosteroid use, it would be important to mention if the authors are talking about systemic steroids or inhaled steroids, or both. It appears they may be talking about systemic (oral or IV) steroids. Very important to make that distinction.
- Line 244- isn’t CNS involvement with MPP actually RMPP? So this sentence should be combined with the one before it, stating the same thing.
- Line 247-258 - length of treatment with corticosteroids is very important. Usual 30 mg/kg/day dose is only for 3 days, and is usually only done for severe, acute inflammatory interstitial lung disease disorders. 2 mg/kg/day dose is not high dose – it is the regular or standard recommended dose, 1mg/kg/dose twice daily for 5 days. A 10 day course of this would be considered “extended steroid course” for most refractory disease states. Anything below this is undertreatment, and should be considered “low dose systemic corticosteroid therapy”. Thus, the length of treatment course
- Line 262 - Is there a standard definition of RMPP? Does it really require at least 7 days or more of macrolide antibiotics? Or is this all only at the author’s institution where this exists? All of this would be new information, as this was not introduced in the Introduction section. Would need to add it there also, in order to discuss this here.
- Line 269 – an increasing IL-18 value in MPP patients was also indicative. Please be specific, as this biomarker is also not specific. Not all high IL-18 levels in patients means it is RMPP.
- Line 270 - This sentence – “Most patients achieved defervescence within 48 hours of initiation of steroid therapy,” - what dose of steroid does this refer to? And interval? Often in the ICU, patients are treated with 1 mg/kg/dose q6 hours before one is considered “steroid resistant”.
- Line 270 – how important is fever as an endpoint in treatment efficacy? Does fever degree or fever length of course correlate with severity of lung findings, or other organ disease findings? Are we treating the fever, or the patient? None of this was discussed. Also, is this fever despite taking around the clock antipyretics (ibuprofen or acetaminophen, etc), or without antipyretics? What is considered fever here?
- Line 273 – any recommendation for IVIG here needs to be carefully considered here. There is no data given as to why IVIG “should be considered”. Is there positive outcome data on it somewhere in regards to RMPP? If so, it should be presented here. The verbiage about IVIG is too strong here without data presentation to back it up.
- Line 277 - what about the use of inhaled steroids? Is that beneficial at all? Should comment on this and show review of literature on this.
- Line 281 – flexible bronchoscopy is not contraindicated in cases of pneumonia at all. Pulmonologist do them all the time to remove airway plugs, or to sample BAL to better tailor antibiotic treatment. Therefore, this statement should be removed.
- Line 282 – I thought RMPP is defined as MPP with extrapulmonary complications, and not MPP with worsening pulmonary disease – wouldn’t that be categorized under Severe MPP (line 133)?
- Line 283 – 293 – Need more information here. Performing a bronchoalveolar lavage is a diagnostic procedure, and not therapeutic. It appears that the authors here suggest that performing a BAL is therapeutic. Do they mean removing plugs from airways to open the airways up? How does infusing saline into alveoli help with pneumonia? Has this been studied? Is there data on this?
- Line 285 – efficacy and safety are two different things, but they are lumped here into one. Efficacy, as measured in terms of what endpoint? Safety in terms of what adverse effect evaluated that was not seen (airway hemorrhage, worsening oxygen requirement, need for intubation, etc)? All this needs to be stated in more detailed terms, if one is to claim that performing BAL improves a patient.
- Line 286 – How is “disease remission” defined in RMPP? Is this improvement of several morbidity endpoints, including what exactly (improved cough, improved chest X-ray, improved respiratory distress, improved LDH, improved CRP – which?)
- Line 291-293 - Looks like the bronchoscopy was actually for airway obstructing plastic bronchitis removal. That is actually considered more a foreign body removal procedure than BAL. BAL is a diagnostic procedure performed after wedging the bronchoscope far into a distal airway, so as to sample the alveolar contents. It is not a removal of a substance such as airway casts. The term needs to be ‘therapeutic bronchoscopic airway plug removal” instead.
- Line 302-304 – This statement is too finite and definite. Use of steroids and IVIG for RMPP is not standard of care. It has shown benefit in some patients, but all it can be said here is that both have been shown to possibly show benefit. And not that their use “has been confirmed”.
Reviewer 3 Report
The article regarding the refractory Mycoplasma pneumoniae pneumonia in children is a special one.
There is a very interesting approach of a new tool for paediatric patients and very useful in clinical practice or clinical research to complete the functional evaluation with early identification of RMPP.
It is a significant contribution in the field, with very good references , very good iconography and high level scientific presentation of the data.
Author Response
Thank you very much for your recognition of our article and we hope it will help some pediatricians and patients in their clinical practice
This manuscript is a resubmission of an earlier submission. The following is a list of the peer review reports and author responses from that submission.